# Differential-Perceptive and Retrieval-Augmented MLLM for Change Captioning

## ABSTRACT

Change captioning involves describing the subtle changes between a pair of similar images. Although existing efforts have achieved compelling success, they overlook the potential of multimodal large language models (MLLMs) in tackling this challenging task. In this work, we aim to empower MLLMs with the capability to perceive subtle differences between paired images and enhance their performance in generating change captions. Specifically, we present a diFferentIal-perceptive aNd rEtRieval-augmented MLLM (FINER-MLLM) tailored for this task. In particular, FINER-MLLM leverages LoRA fine-tuned MLLM's image encoder to extract image patch features, enabling the capture of detailed image information. Subsequently, within MLLM's feature extraction, typically Q-Former, FINER-MLLM incorporates dual constraints: the intra-image feature independence constraint and the inter-image feature alignment constraint. These constraints ensure that the features can comprehensively extract subtle visual information within each image and that corresponding features across images align effectively. Last, we introduced the retrieval augmentation to first retrieve the relevant corpus to facilitate the MLLM's decoder *i.e.*, LLM, in generating accurate change captions. Extensive experiments on three benchmark datasets, *i.e.*, CLEVR-Change, Spot-the-Diff, and Image-Editing-Request, demonstrate the superiority of our proposed method.

## CCS CONCEPTS

• **Computing methodologies** → *Natural language generation*; *Image representations*.

## KEYWORDS

Change captioning, Retrieval-augmented captioning, Multimodal large language model

### ACM Reference Format:

Anonymous Author(s). 2024. Differential-Perceptive and Retrieval-Augmented MLLM for Change Captioning. In *Proceedings of Make sure to enter the correct conference title from your rights confirmation emai (mm'24)*. ACM, New York, NY, USA, 9 pages. https://doi.org/10.1145/nnnnnnn.nnnnnnn

## 1 INTRODUCTION

Unlike conventional image captioning [27], which focuses solely on understanding the content of a single image, change captioning is

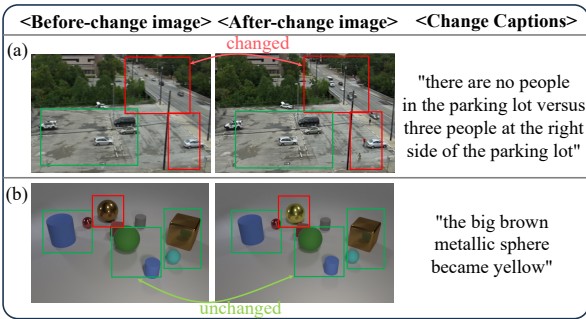

**Figure 1: Cases of change captioning. The green/red boxes denote the unchanged/changed visual details, respectively.**

a more challenging task of vision and language. This task requires not only comprehending the contents of two similar images but also describing their differences using natural language. Given its great potential value in various real-world applications, such as anomaly detection [9, 20] and pathological change report [18, 21], change captioning has garnered increasing research attention in recent years.

Although existing efforts have shown promising performance in change captioning, none have explored the application of advanced multimodal large language models (MLLMs) [2, 7, 17, 42] in this task. In mainstream MLLM paradigms such as BLIP-2 [17] and InstructBLIP [7], a pivotal strategy involves utilizing a learnable Q-Former [17] to integrate a pre-trained frozen image encoder and a frozen large language model (LLM). The learnable Q-Former introduces a set of queries to capture fine-grained image features, aligning them effectively with the semantic space of the LLM. Consequently, both the image encoder and LLM do not require further fine-tuning yet exhibit remarkable success in various multimodal tasks necessitating both visual comprehension and text generation. With this in mind, our work aims to leverage MLLMs to harness the power of MLLMs to tackle the challenging task of change captioning more effectively.

Essentially, the key to the change captioning lie in two points: (1) extracting visual features from two images to perceive subtle visual differences, and (2) generating natural language descriptions for these differences. However, directly applying existing MLLMs to change captioning presents challenges for both of these points.

For visual feature extraction, when employing existing MLLMs directly, the image undergoes processing through a frozen visual encoder and a learnable Q-Former, which generate a set of visual queries. However, some of the learned visual queries may focus on the same prominent information in the image, resulting in redundancy and the inability to adequately capture subtle visual details. For example, subtle visual details, such as the pedestrians highlighted within the red boxes in Figure 1(a), are prone to change and may be overlooked due to their small size, leading to an incomplete representation of the image's content. Furthermore, when

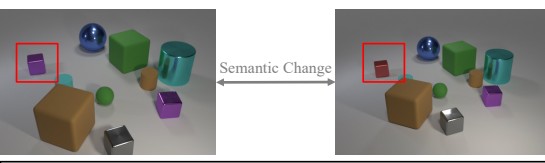

Retrieval relevant corpus:
the small purple metal cube behind the brown cylinder changed to red.
the tiny purple metal cube that is behind the small brown thing changed to red.
the tiny purple shiny block that is behind the large blue matte object changed to red.
the tiny purple shiny block that is left of the brown cylinder turned red.
the small purple shiny cube behind the red thing changed to red.

**Figure 2: Semantically relevant sentences in change captioning. Relevant phrases are highlighted in green.**

MLLMs are applied to extract image features separately from the two images including before-change image and after-change image, the two forward propagation processes may not align features of the same visual query to the same regions within the images. This misalignment is problematic because changes often occur in the same region of both images, as shown in Figure 1(b), where the color of the ball changes at the same position. Accordingly, the inability to consistently map the same image regions into same visual queries hinder the model's effectiveness in perceiving differences. Therefore, **how to enhance the capability of existing MLLMs to comprehensively extract subtle visual information from images and ensuring that the same visual queries correspond to the same regions in both input images poses the first challenge**.

As for differential description generation, the LLM in MLLMs has predominantly pre-trained on complete image descriptions, with limited exposure to differential descriptions. Employing the LLM without fine-tuning, as commonly done in existing methods, would result in suboptimal performance. However, fine-tuning the LLM would incur high computational costs and is prone to overfitting due to its large parameter size. Consequently, **how to enhance the performance of LLM in generating differential descriptions in a cost-effective manner is also a crucial challenge**.

To tackle the aforementioned challenges, we aim to refine the differential-perception capabilities within the feature extraction process of paired images. Additionally, we endeavor to develop a retrieval-augmented methodology to enhance the generation of differential descriptions. We present a di**F**ferent**I**al-perceptive a**N**d r**Et**rieval-augmented MLLM (FINER-MLLM) for change captioning. Specifically, FINER-MLLM employs InstructBLIP as the backbone and consists of three key components: image encoding, differential-perceptive feature modeling, and retrieval-augmented caption generation. The first component involves extracting image patch features with LoRA fine-tuning of the image encoder. The second component focuses on extracting paired image features through Q-Former, where we introduce dual constraints to enhance the differential perception capabilities. Specifically, we devise an intra-image feature independence constraint to eliminate redundancy across different visual queries, ensuring that each subtle visual detail is captured. Additionally, we propose an inter-image feature alignment constraint to ensure that corresponding regions in the two input images are mapped to the same visual queries. The third component aims to enhance the generation of differential descriptions by leveraging the assistance of relevant corpus retrieval.

Our motivation stems from the observation that many relevant sentences share highly similar semantics suitable for change captioning. As illustrated in Figure 2, we employ an off-the-shelf model that utilizes the two images as search queries, providing the top-5 relevant sentences. Notably, the semantic content of the text highlighted in green accurately describes the difference between the two images. Therefore, we adopt a retrieval-augmented approach, first retrieving the relevant corpus as assistance, and then generating the differential descriptions.

The contribution of this work can be summarized in three folds:

- We are the first to employ MLLMs in the task of change captioning, where we devise dual inter- and intra-feature constraints to enhance the differential perception capabilities in visual feature extraction.
- To the best of our knowledge, we are the first to leverage semantically relevant corpus to enhance the generation of differential descriptions by LLMs in the context of change captioning.
- Extensive experiments conducted on three common datasets validate the superiority of our model. As an additional benefit, we have released our codes to facilitate other researchers[1].

## 2 RELATED WORK

Our work is closely in line with the studies on change captioning, Multimodal Large Language Models and multimodal retrieval-augmented methods.

### 2.1 Change Captioning

Change captioning is a vital task in vision-language understanding and generation [5, 14, 20, 29, 34, 40]. Jhamtani et al. [14] initially created this task from image captioning and change detection, and they proposed a model that aligns clusters of pixel-wise difference between image pairs to output sentences. Park et al. [25] introduced a synthetic dataset including samples with viewpoint distractors. They proposed a dual attention mechanism to localize changed object, along with a dynamic speaker that adaptively attends to visual features at each decoding step. Shi et al. [28] proposed a viewpoint-adapted match encoder to capture both changed and unchanged regions by computing feature-level similarity. They introduced a reinforcement attention method to improve the generation process. Sun et al. [29] proposed a bidirectional difference localization method to locate changes by encoding visual features from two directions. To further enhance the performance of change captioning, existing efforts have been combined with additional tasks [1, 10, 12, 38]. Mehrdad et al. [12] introduced a auxiliary composed image retrieval task to improve the training of change caption network, where these two tasks reinforce each other in a unified framework. Another typical example is IDC-PCL [38], which designs three self-supervised task and contrastive learning strategies to align visual differences and text descriptions at a fine-grained level. Following that, Guo et al. [10] adopted CLIP and the image-text retrieval task for adaption training, which aligns differences in image pairs based on textual descriptions. Different from the previous work, we aim to advance the change captioning task with MLLMs by empowering MLLMs with the ability to perceive subtle

---

[1]https://anonymous.4open.science/r/FINER-MLLM-7010/.

differences between paired images and generate accurate change captioning.

## 2.2 Multimodal Large Language Models

Large language models (LLMs) have demonstrated remarkable abilities in various tasks, but they are primarily designed for text-based inputs. Recently, several multimodal large language models (MLLMs) are proposed to incorporate visual information with LLMs, enabling a fusion of visual and textual modalities for improved performance and understanding. Flamingo [2] introduced a perceiver resampler to combine a frozen visual encoder with LLMs, aligning vision and textual modalities. BLIP2 [17] bridged the modality gap with a lightweight Querying Transformer that transforms visual information to textual query embeddings. LLaVa [22] and MiniGPT-4 [42] both employed the fine-tuning approach with well-collected data, resulting in a significant boost in performance. InstructBLIP [7] further improved the instruction-following ability of models by fine-tuning on multi-modal instruction following datasets. However, existing MLLMs lack the capability to inspect differences among multiple images due to the inability to adequately capture subtle visual details and misalignment of features. Our model is devised to refine the differential-perception capabilities of MLLMs for change captioning.

## 2.3 Multimodal Retrieval-Augmented Methods

Retrieval-augmented methods mainly consist of conditioning generation on additional information that is retrieved from an external datastore [16], which has proven to be effective plugins for both LLMs and MLLMs [11, 27]. Lewis et al. [16] initially proposed retrieval augmented generation for pre-trained language model, which effectively improved its performance on downstream knowledge-intensive tasks. Ramos et al. [27] proposed an image captioning model augmented with retrieval, which is lightweight-training and effective. Specially, they integrated vision and language model by cross attention layer and utilized retrieved captions as prompt to instruct language model, leading to an excellent performance. Zhang et al. [41] proposed a open-book video captioning model that retrieved relevant sentences as reference for better generation. While retrieval augmentation has gained traction in other tasks, it remains largely unexplored in the context of image change captioning [27, 39]. Inspired by retrieval augmentation, our method is to employ a retrieval-then-generation fashion in the context of image change captioning, which utilized a emphasize-mechanism on retrieved corpus to generate accurate sentences.

## 3 METHODOLOGY

In this section, we first formulate the research problem and then detail our proposed method.

## 3.1 Problem Formulation

In this work, we aim to address the challenging change captioning task, which can be formally defined as given a pair of images consisting of a before-change image and an after-change image, the goal is to generate the description of modification between this image pair. Suppose we have a set of triplets, denoted as $\mathcal{D} = \left\{ (I_{bef}, I_{aft}, T_{diff})_i \right\}_{i=1}^{N}$, where $I_{bef}$, $I_{aft}$, and $T_{diff}$ refer to

the before-change image, the after-change image, and the caption of difference, respectively. $N$ is the total number of triplets. Following the traditional pipeline of retrieval-augmented methods, we first utilize a retriever to obtain semantically relevant corpus $T_{ref}$ as reference knowledge, then we train the generator to produce caption $T_{diff}$ conditioned on retrieval descriptions. The overall process is formally given as follows,

$$p(y \mid I_{bef}, I_{aft}) = \sum_{T_{ref} \subset \mathcal{D}} \underbrace{p(T_{ref} \mid \mathcal{R}(I_{bef}, I_{aft}))}_{\text{Retriever}} \cdot \underbrace{p(y \mid \mathcal{G}(I_{bef}, I_{aft}, T_{ref}))}_{\text{Generator}},$$ (1)

where $y$ is the tokens of $T_{diff}$, $\mathcal{R}$ represents the function of retriever, and $\mathcal{G}$ denotes the function of generator.

## 3.2 FINER-MLLM

As a major novelty, we propose the extraction of differential-perceptive features from image pairs and adopt a retrieval-augmented paradigm to generate more precise change captions. Specifically, we propose a Differential-Perceptive and Retrieval-Augmented MMLM (FINER-MLLM), which is illustrated in Figure 3. It consists of three key components: (a) image encoding, (b) differential-perceptive feature extraction, and (c) retrieval-augmented caption generation. The first module works on extracting image patch features of the input image pair. In this module, LoRA fine-tuning is applied to the image encoder, which significantly reduces the training overhead compared to full-parameter fine-tuning. The second module targets extracting differential-perceptive query features by Q-Former, where the attention distributions over image patch features of query between before-change image and after-change image are well-aligned, combined with comprehensive subtle visual information extraction. The third module aims to optimize the generation of LLM by retrieval augmentation. Specifically, we first retrieve semantically relevant corpus from the training set based on the input image pair, then highlight the prediction weight of the right words in retrieved sentences to improve the prediction of frozen LLMs. We now elaborate on each module of FINER-MLLM.

## 3.3 Image Encoding

Following [7], we adopt EVA-ViT-g/14 [8] as the image encoder, which has shown remarkable success in various vision tasks. We aim to get aligned visual queries through attention constraints. To fulfill this, we need to first obtain the fine-grained features of before-change image and after-change image, and then regularize the attention of visual queries to get aligned queries for accurate difference inference. However, the off-the-shelf image encoder limits its ability to provide differentiated fine-grained features for similar images, thus inhibiting the learning of cross-modal visual queries in Q-Former. Toward this end, we propose to add parallel lightweight low-rank adapters (LoRA) in the frozen Vision Transformer, to extract image patch features of similar images.

The core of LoRA is to efficiently fine-tune pre-trained large models by introducing low-rank matrices into the model's parameters.

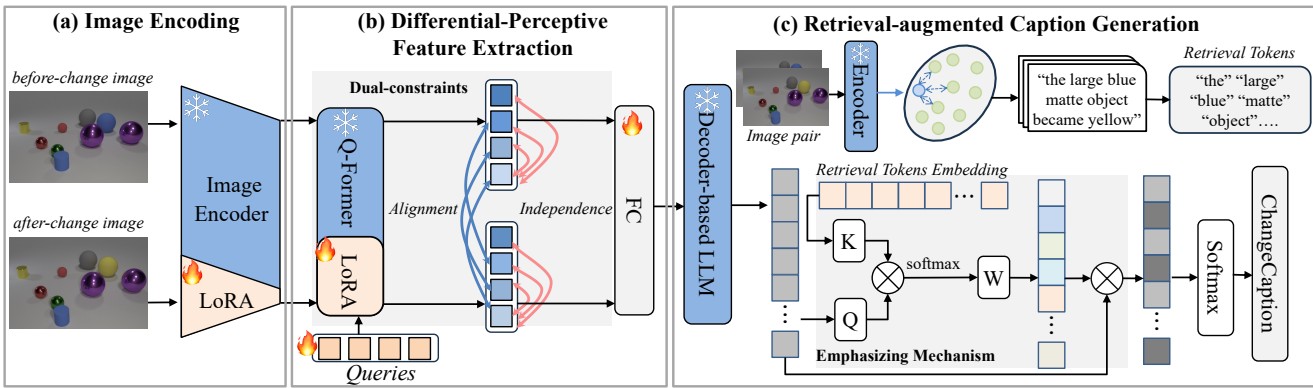

**Figure 3: The proposed FINER-MLLM consists of three key modules: (a) image encoding, (b) differential-perceptive feature extraction, and (c) retrieval-augmented caption generation.**

Here we integrate parallel low-rank matrices into specific linear layers within Transformer blocks to fine-tune the pre-trained image encoder for change captioning. For clarity, here we use the fine-tuning process of a specific linear layer with weight matrix $\mathbf{W} \in \mathbb{R}^{U \times K}$ as an example to introduce the LoRA fine-tuning technique. Specifically, LoRA applies a low-rank decomposition by projection-down and projection-up weights, which is formulated as follows,

$$\mathbf{h} = \mathbf{Wx} + \Delta \mathbf{Wx} = \mathbf{Wx} + \alpha \mathbf{BAx}, \qquad (2)$$

where $\mathbf{x} \in \mathbb{R}^K$ and $\mathbf{h} \in \mathbb{R}^U$ refer to the input and output features, respectively. $\mathbf{A} \in \mathbb{R}^{R \times K}$ is the projection-down weight, $\mathbf{B} \in \mathbb{R}^{U \times R}$ is the projection-up weight, $R$ is the rank size $(R \ll \min(U, K))$, and $\alpha > 0$ is a scaling hyper-parameter. In this way, the full-parameter fine-tuning of optimizing the matrix $\Delta \mathbf{W}$ is replaced by optimizing the decomposed low-rank matrices $\mathbf{B}$ and $\mathbf{A}$, which can greatly reduce the number of parameters to be optimized. Generally, there are four weight matrices in the self-attention module $(\mathbf{W}_q, \mathbf{W}_k, \mathbf{W}_v, \mathbf{W}_o)$ and two in the feed-forward layer within the Transformer block. Here, we insert LoRA adapters to optimize $\mathbf{W}_q$ and $\mathbf{W}_v$ in the self-attention module, as well as two weight matrices in the feed-forward layer.

Given an input image pair comprising a before-change image and an after-change image, we can extract their image patch features using the LoRA image encoder, respectively, which is as follows:

$$\mathbf{X}_i = \left\{ I_i^1, \cdots I_i^N \right\}, i \in \{bef, aft\}, \qquad (3)$$

where $I_i^n \in \mathbb{R}^D$ $(n = 1, \cdots, N)$ denotes the local image patch feature, $D$ is feature dimension, $N$ is the number of image patches.

### 3.4 Differential-Perceptive Feature Extraction

After obtaining the image patch features of the before-change image and the after-change image, we feed them into Q-Former to extract query features, respectively. The Q-Former in InstructBLIP [7] utilizes a fixed number of learnable query vectors to interact with image path features through cross-attention layers for aggregating visual information. Additionally, the query vectors interact with each other through self-attention layers, enabling the sharing of visual information among them. When applied in the change captioning task, the query vectors individually interact with every

single image of pairs, which presents two issues: (1) The query features extracted from a single image may contain redundancy, which can not obtain comprehensive visual information of the image and hence lose certain details. However, these visual details generally include changed visual information that needs to be focused on change captioning. (2) The query vectors extract features from two images independently, where the features between image pairs in the same location may focus on different visual regions. This discrepancy makes LLMs fail to accurately perceive visual changes based on the extracted features. Hence, it is necessary to eliminate redundancy across different visual queries and align visual regions of the corresponding query of image pairs. Toward this end, we design dual constraints as shown in Figure 4: (1) intra-image feature independence constraint and (2) inter-image feature alignment constraint. The former intra-image feature independence constraint ensures that queries attend to different image patch features within each image pair, enhancing the extraction of comprehensive visual details. The latter inter-image feature alignment constraint ensures that corresponding queries attend to the same visual regions across image pairs, facilitating interaction between the images to identify similarities and differences.

Specifically, we denote the query features extracted from the image patch features as $Q^i = \{\mathbf{q}_1^i, \mathbf{q}_2^i, \cdots \mathbf{q}_K^i\}$, where $i \in \{bef, aft\}$ denotes the before-change and after-change image, respectively. $K$ is the number of query features and $\mathbf{q}_k^i \in \mathbb{R}^D$ denotes the $k$-th query feature. When obtaining $\mathbf{q}_k^i$, the cross-attention weight distribution[2] for aggregating image patch features is denoted as $\mathbf{p}_j^i = [p_1, p_2, \cdots, p_N] \in \mathbb{R}^N$, where $p_n$ is a cross-attention weight scalar for aggregating the $n$-th image path feature. The cross-attention weight distributions of the $K$ query features are denoted as $\mathcal{A}^i = \{\mathbf{p}_1^i, \mathbf{p}_2^i, \cdots, \mathbf{p}_K^i\}$.

**Intra-image Feature Independence Constraint.** We employ the commonly used InfoNCE [36] loss as the orthogonal loss for extracting comprehensive visual details. Essentially, the InfoNCE loss aims to enforce the identical distributions close to each other and dissimilar distributions away from each other, namely, minimizing the similarity between different distributions. Therefore, we apply

---

[2]To simplify the explanation, we default to using a single attention head for illustration purposes.

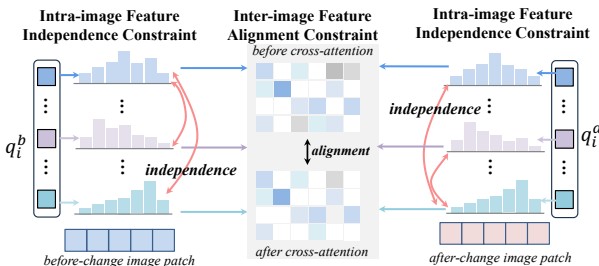

**Figure 4: Differential-Perceptive Feature Extraction.**

this loss to the cross-attention distributions of each query feature, respectively. Mathematically, we have the following loss function for optimizing the extraction of semantic features as follows,

$$\mathcal{L}_{ortho} = \frac{1}{K} \sum_{k=1}^{K} - \log \left\{ \frac{\exp \left\{ s \left( \mathbf{p}_k^i, \mathbf{p}_k^i \right) / \tau \right\}}{\sum_{b=1}^{B} \exp \left\{ s \left( \mathbf{p}_k^i, \mathbf{p}_b^i \right) / \tau \right\}} \right\}, \quad (4)$$

where $B$ is the batch size, $s(\cdot, \cdot)$ denotes cosine similarity function, and $\tau$ is the temperature factor.

**Inter-image Feature Alignment Constraint.** As mentioned above, each image has $K$ cross-attention distributions through Q-Former, denoted as $\mathcal{A}^{bef}$ and $\mathcal{A}^{aft}$. These distributions manifest the visual regions targeted by the query features. To encourage the query features to concentrate on identical regions, we design the inter-image feature alignment constraint with the Kullback-Leibler (KL) Divergence between $\mathcal{A}^{bef}$ and $\mathcal{A}^{aft}$ as follows,

$$D_{KL} \left( \mathbf{p}_1 \| \mathbf{p}_2 \right) = \sum_{n=1}^{N} p_n^1 \log \frac{p_n^1}{p_n^2}, \quad (5)$$

$$\mathcal{L}_{con} = \frac{1}{K} \sum_{k=1}^{K} \left\{ D_{KL} \left( \mathbf{p}_k^{bef} \| \mathbf{p}_k^{aft} \right) + D_{KL} \left( \mathbf{p}_k^{aft} \| \mathbf{p}_k^{bef} \right) \right\}. \quad (6)$$

Note that we also integrate LoRA adapters into the Q-Former model instead of conducting full-parameter fine-tuning. In particular, we insert low-rank matrices into the query projection and value projection layer in the self-attention and cross-attention layer of Q-Former.

## 3.5 Retrieval-augmented Caption Generation

The extracted query features are concatenated and fed into the fully connected layer to map the dimensions the same as the input of LLM. Then, the query features are fed into LLM to generate change captions. As aforementioned above, many relevant sentences in the dataset contain potential descriptions of similar change scenes or the same object-level phrases. Therefore, we resort to retrieval-augmented caption generation to utilize the potentially useful information in the relevant sentences. We initially introduce a visual-to-text retriever to retrieve semantically relevant sentences from training data based on input image pairs. To effectively utilize the retrieved sentences, we design the retrieval-augmented caption generation module at the word level. Specifically, we employ a novel emphasize mechanism on individual words within the retrieved sentences, which aims to copy the useful words in sentences by prioritizing words directly.

**Caption Decoder.** The LLM predicts the right word based on semantic features of image pairs $[Q_{bef}; Q_{aft}]$ at each time step by computing the current hidden state $h_t^c$ in parallel. The current hidden state of LLM $h_t^c$ depends on the previous hidden states and input semantic features of images. Then LLM generates the probability distribution of the fixed vocabulary $\mathbf{p}_v$ through a fully connected layer at each time step:

$$\mathbf{p}_v = softmax(\mathbf{W}_v h_t^c + \mathbf{b}_v), \quad (7)$$

where $\mathbf{p}_v \in \mathbb{R}^{1 \times V}$, $\mathbf{W}_v$ and $\mathbf{b}_v$ are frozen parameters of last output layer in LLM.

**Generation Augmentation.** We first utilize retriever to get the top-$k$ most relevant sentences based given image pair. Each retrieved sentence consists of a set of words with their embeddings. We aggregate these sentences into one embedding set as retrieved vocabulary by removing repeated words, denoted as $y = \{y^1, \cdots, y^L\}$. At each decoding step $t$, the module acts on retrieved vocabulary, uses the current hidden state $h_t^c$ as the query to attend retrieved words, and produces the word probability distribution over retrieved vocabulary $\mathbf{p}_r \in \mathbb{R}^{1 \times L}$ of the corresponding sentence, which is formulated as follows:

$$\mathbf{p}_r, \mathbf{c}_r = Att(\mathbf{h}_t^c, y, y; \theta_r), \quad (8)$$

where $Att(\cdot)$ is the additive attention module with parameters of $\theta_r$, $\mathbf{p}_r$ denotes the relevant degree between words in retrieved vocabulary and current hidden state, and $\mathbf{c}_r$ represents the context of the retrieved words that is the weighted summation of $y$ by $\mathbf{p}_r$. Then, we apply a emphasize mechanism to highlight relevant words over $\mathbf{p}_v$ at each time step. Formally, we prioritize the words by adding weight to the corresponding value in the probability distribution over fixed vocabulary:

$$\mathbf{p}_c = \mathbf{p}_v(1 + \mathbf{p}_r), \quad (9)$$

where $\mathbf{p}_r \in \mathbb{R}^{1 \times L}$ is extended to the same size of fixed vocabulary. In this manner, only the prediction of relevant words is improved, without altering remaining values in the prediction. The goal of generation is to minimize the negative log-likelihood of each target word $y_t$:

$$\mathcal{L}_{gen} = - \sum_{t=1}^{T} y_t \log \mathbf{p}_{ct}, \quad (10)$$

*Optimization.* Integrating the three key modules, we parameterize the final objective function for our model as follows,

$$\Theta^* = \arg \min_{\Theta} \left( \mathcal{L}_{gen} + \lambda \mathcal{L}_{ortho} + \eta \mathcal{L}_{con} \right), \quad (11)$$

where $\Theta$ denotes the to-be-learned parameters in FINER-MLLM, $\lambda$ and $\eta$ are the trade-off hyper-parameters.

## 4 EXPERIMENT

In this section, we first introduce the experimental settings and then provide the experiment results and corresponding analyses

## 4.1 Experimental Settings

*4.1.1 Datasets.* We chose three public datasets to evaluate our model, including a synthetic dataset CLEVR-Change [25], as well as two real-world domain datasets Spot-the-Diff [14] and Image-Editing-Request [30].

**CLEVR-Change** [14] is a synthetic dataset that includes large-scale scene changes of geometry objects and distractors from viewpoint variations. It contains $67,660$, $3,976$ and $7,970$ image pairs for training, validation and testing, respectively. Following [25] we utilized the official dataset split protocol to validate our method.

**Spot-the-Diff** [25] is real-world dataset collected from video-surveillance footage, consisting of $13,192$ image pairs captured from a fixed viewpoint at different timestamps. Following previous work [25], we evaluated our model in a single caption setting and split image pairs into 80% for training, 10% for validation and 10% for testing.

**Image-Editing-Request** [30] consists of $3,939$ image pairs from real life with $5,695$ editing instructions. Following [33], we used the official dataset split protocol with $3,061$ image pairs for training, 383 image pairs for validation, and 495 image pairs for testing.

*4.1.2 Implementation Details.* Our model is based on Instruct-BLIP [7], which includes a ViT backbone and a Q-Former to bridge a frozen decoder-based large language model. Following BLIP2 [17], we adopted EVA-ViT-g/14 [8] and Vicuna-7B [6] as the image encoder and large language model, respectively. We trained model by AdamW optimizer [23] with weight decay of 0.1. We applied a cosine learning rate decay with a fixed peak learning rate and a fixed number of linear warm-up steps. For CLEVR-Change, the peak learning rate and warm-up steps are set to $5e-5$ and $4,000$, respectively. For Spot-the-Diff, the peak learning rate and warm-up steps are set to $3e-5$ and $1,000$, respectively. For Image-Editing-Request, the peak learning rate and warmup steps are set to $3e-5$ and 250, respectively. The batch size is set to 16 for all datasets. The temperature factor $\tau$ in Eqn. (4) is set to 0.1. Regarding the trade-off hyper-parameters, we set $\lambda = 0.25$, $\eta = 1.0$ for CLEVR-Change, $\lambda = 0.75$, $\eta = 1.0$ for Spot-the-Diff and $\lambda = 0.75$, $\eta = 0.1$ for Image-Editing-Request. The rank size of low-rank matrices in image encoder is set to 16 for CLEVR-Change, 4 for Spot-the-Diff, and 4 for Image-Editing-Request, respectively. The rank size of low-rank matrices in Q-Former is set to 4 for all datasets. The retrieval augmented generation requires a effective retriever for visual-to-text retrieval. We introduced the pre-trained retrieval model from CLIP4IDC, which is trained to retrieve corresponding change captions based on image pairs. We utilized this model to retrieve top-200 semantically relevant sentences from texts in train set. We utilized top-5 sentences for augmented generation for all datasets. We re-implemented the retrieval model on Image-Editing-Request based on CLIP4IDC, due to the fact that they do not provided pre-trained weights. All the experiments are implemented by PyTorch over a server equiped with 8 A100 GPUs, and the random seeds are fixed for reproducibility.

*4.1.3 Evaluation.* We used the standard evaluation protocol for each dataset and reported the same metrics for a fair comparison. For change captioning, we reported five standard metrics including BLEU-4(B) [24], METEOR(M) [4], ROUGE-L(R) [19], CIDEr-D(C) [37] and SPICE(S) [3]. Following [10], we compute the metrics by Microsoft COCO evaluation server.

**Table 1: Performance comparison on CLEVR-Change with respect to all metrics. The best results over baselines are underlined, while the overall best results are in boldface.**

| Method | B | M | R | C | S |
|---|---|---|---|---|---|
| DUDA [25] (CVPR'19) | 47.3 | 33.9 | – | 112.3 | 24.5 |
| VAM [28] (ECCV'20) | 50.3 | 37.0 | 69.7 | 114.9 | 30.5 |
| VAM+ [28] (ECCV'20) | 51.3 | 37.8 | 70.4 | 115.8 | 30.7 |
| IFDC [13] (TMM'21) | 49.2 | 32.5 | 69.1 | 118.7 | – |
| DUDA+Aux [12] (CVPR'21) | 51.2 | 37.7 | 70.5 | 115.4 | 31.1 |
| VACC [15] (ICCV'21) | 52.4 | 37.5 | – | 114.2 | 31.0 |
| SRDRL [35] (ACL'21) | 54.9 | 40.2 | 73.3 | 122.2 | 32.9 |
| $R^3$Net [34] (EMNLP'21) | 54.7 | 39.8 | 73.1 | 123.0 | 32.6 |
| BiDiff [29] (Int. J. Intell'22) | 54.2 | 38.3 | – | 118.1 | 31.7 |
| IDC-PCL [38] (AAAI'22) | 51.2 | 36.2 | 71.7 | 128.9 | – |
| CLIP4IDC [10] (AACL'22) | **56.9** | 38.4 | **76.4** | 150.7 | – |
| NCT [32] (TMM'23) | 55.1 | 40.2 | 73.8 | 124.1 | 32.9 |
| VARD [31] (TIP'23) | 55.2 | 40.8 | 74.1 | 124.1 | 33.3 |
| SCORER [33] (ICCV'23) | 56.3 | **41.2** | 74.5 | 126.8 | **33.3** |
| **FINER-MLLM** | 55.6 | 36.6 | 72.5 | 137.2 | 26.4 |

**Table 2: Performance comparison on Spot-the-Diff with respect to all metrics. The best results over baselines are underlined, while the overall best results are in boldface.**

| Method | B | M | R | C | S |
|---|---|---|---|---|---|
| DDLA [14] (EMNLP'18) | 8.5 | 12.0 | 28.6 | 32.8 | – |
| DUDA [25] (CVPR'19) | 8.1 | 11.8 | 29.1 | 32.5 | – |
| VAM [28] (ECCV'20) | 10.1 | 12.4 | 31.3 | 38.1 | – |
| VAM+ [28] (ECCV'20) | 11.1 | 12.9 | 33.2 | 42.5 | 17.1 |
| IFDC [13] (TMM'21) | 8.7 | 11.7 | 30.2 | 37.0 | – |
| DUDA+Aux [12] (CVPR'21) | 8.1 | 12.5 | 29.9 | 34.5 | – |
| VACC [15] (ICCV'21) | 9.7 | 12.6 | 32.1 | 41.5 | – |
| SRDRL [35] (ACL'21) | – | 13.0 | 31.0 | 35.3 | 18.0 |
| $R^3$Net [34] (EMNLP'21) | – | 13.1 | 32.6 | 36.6 | 18.8 |
| BiDiff [29] (Int. J. Intell'22) | 6.6 | 10.6 | 29.5 | 42.2 | – |
| CLIP4IDC [10] (AACL'22) | 11.6 | 14.2 | 35.0 | 47.4 | – |
| VARD [31] (TIP'23) | – | 12.5 | 29.3 | 30.3 | 17.3 |
| SCORER [33] (ICCV'23) | 10.2 | 12.2 | – | 38.9 | 18.4 |
| **FINER-MLLM** | **12.9** | **14.7** | **35.5** | **61.8** | **22.1** |

**Table 3: Performance comparison on Image-Editing-Request with respect to all metrics. The best results over baselines are underlined, while the overall best results are in boldface. † denotes the model does not utilize the retrieved relevant corpus in caption generation.**

| Method | B | M | R | C | S |
|---|---|---|---|---|---|
| DUDA [25] (CVPR'19) | 6.5 | 12.4 | 37.3 | 22.8 | – |
| BiDiff [29] (Int. J. Intell'22) | 6.9 | 14.6 | 38.5 | 27.7 | – |
| CLIP4IDC [10] (AACL'22) | 8.2 | 14.6 | 40.4 | 32.2 | – |
| NCT [32] (TMM'23) | 8.1 | 15.0 | 38.8 | 34.2 | 12.7 |
| VARD [31] (TIP'23) | 10.0 | 14.8 | 39.0 | 35.7 | – |
| SCORER [33] (ICCV'23) | 10.0 | 15.0 | 39.6 | 33.4 | – |
| **FINER-MLLM** | 13.3 | 14.6 | 39.6 | 50.5 | 14.0 |
| **FINER-MLLM†** | **14.1** | **15.9** | **40.4** | **53.3** | **15.9** |

## 4.2 Performance Comparison

To validate the effectiveness of our method in the context of change captioning, we compared FINER-MLLM with the following baselines: DDLA [14], DUDA [25], VAM [28], IFDC [13], VACC [15], SRDRL [35], $R^3$Net [34], BiDiff [29], IDC-PCL [38], NCT [32], VARD [31], SCORER [33], and CLIP4IDC [10]. Among them, DDLA [14] initially created this task. It aligns clusters of differential pixels between image pairs to generate change captions. DUDA [25] is the first work to address viewpoint variations, which utilizes dual attention to locate visual changes and a dynamic speaker to generate change captions. CLIP4IDC [10] adopts CLIP [26] as the image feature extraction backbone and designs a two-stage approach including pre-training and fine-tuning to capture differences in image pairs.

Tables 1, 2 and 3 summarize the performance comparison on the three datasets. The missing results of some methods are because they do not report their results on the respective datasets. From these tables, we obtained the following observations: 1) FINER-MLLM consistently outperforms all baseline methods across the real-world domain datasets, including Spot-the-Diff and Image-Editing-Request. This confirms the advantage of our model of employing differential-perceptive and retrieval-augmented MLLM in the context of change captioning. 2) We also noticed that our FINER-MLLM exhibits inferior performance on the CLEVR-Change dataset. This may be attributed to two reasons. Firstly, the images in CLEVR-Change are all synthesized geometries and are not from real-world data. The pre-training datasets of MLLM may contain only a few synthesized geometrical images, leading to the poor generalization ability of MLLM on this dataset. Secondly, the CLEVR-Change dataset includes changes in viewpoint with a high percentage (around 60%), such as slight rotations or scaling of the image, whereas the other two real-world scene datasets hardly contain changes in viewpoint. Through our designed differential-perceptive feature modeling, we reduced the model's capability to perceive changes in viewpoint, resulting in poor performance in the synthesized dataset. While we want to claim that in real-world scenarios, change captioning often does not involve changes in viewpoint, such as surveillance cameras and medical image lesion comparisons. This further highlights the potential application values of our FINER-MLLM in real-world scenarios. 3) We also observed that in the Image-Editing-Request dataset, the quality of the retrieved relevant corpus is not as good as the other two datasets. Accordingly, we reported the results of FINER-MLLM†, which does not utilize the retrieved relevant corpus in text generation. It can be seen from Table 3 that FINER-MLLM† yields slightly better performance compared to FINER-MLLM. This may be due to the diverse nature of change captions in this dataset, where the retrieved relevant corpus cannot provide assistance and may even confuse the LLM.

## 4.3 Ablation Study

To verify the importance of each component in our model, we compared FINER-MLLM with its following derivatives.

- **w/o_LoRA-ViT**: To explore the necessity of fine-tuning the image encoder, we froze the parameters of EVA-ViT-g/14.

**Table 4: Ablation study on Spot-the-Diff.**

| Method | B | M | R | C | S |
|---|---|---|---|---|---|
| w/o_LoRA-ViT | 12.0 | 13.3 | 33.9 | 59.9 | 21.6 |
| w/o_Dual-constraints | 9.7 | 11.4 | 32.3 | 56.3 | 20.8 |
| w/o_Independence | 12.3 | 14.4 | 35.0 | 59.1 | 21.8 |
| w/o_Alignment | 11.9 | 13.6 | 34.3 | 64.1 | 23.2 |
| w/o_RAG | 12.1 | 14.0 | 34.6 | 60.1 | 21.0 |
| **FINER-MLLM** | **12.9** | **14.7** | **35.5** | **61.8** | **22.1** |

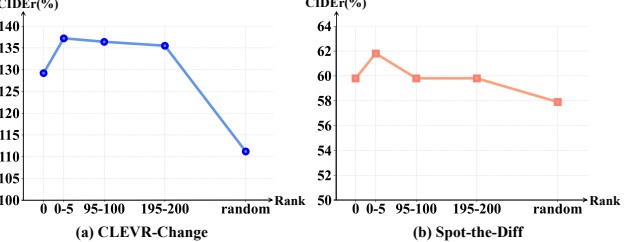

**Figure 5: Influence of different settings of retrieved sentences on (a) CLEVR-Change and (b) Spot-the-Diff.**

- **w/o_Dual-constraints**: To investigate the importance of differential-perceptive capabilities, we removed both the intra-image feature independence constraint and inter-image feature alignment constraint by setting $\lambda = 0$ and $\eta = 0$ in Eqn. (11).
- **w/o_Independence**: To further check the effect of feature independence, we directly set $\lambda = 0$ in Eqn. (11).
- **w/o_Alignment**: To gain more insight into the influence of feature alignment, we set $\eta = 0$ in Eqn. (11).
- **w/o_RAG**: To study the effect of the retrieval-augmented approach, we discard the retrieved relevant corpus in the caption generation process.

Table 4 shows the ablation results of FINER-MLLM on Spot-the-Diff. From this table, we gained the following observations: 1) w/o_LoRA-ViT is inferior to FINER-MLLM, which proves that fine-tuning the image encoder is beneficial for providing image patch features with more detailed visual information for the latter differential-perceptive feature extraction. 2) w/o_Dual-constraints yields the worst performance, while both w/o_Independence and w/o_Alignment perform inferior to FINER-MLLM. This highlights the importance of differential-perceptive capabilities for change captioning in real-world scenarios. The designed dual constraints ensure both intra-image feature independence and inter-image feature alignment. 3) FINER-MLLM outperforms w/o_RAG, which confirms the benefit of utilizing retrieved relevant corpus to effectively enhance the change caption generation process.

To delve deeper into retrieval-augmented caption generation, we examined the impact of the quality of the retrieved relevant corpus on FINER-MLLM's performance, as depicted in Figure 5. The x-axis represents the utilized relevant corpus, with ranking results of the retrieved relevant corpus sorted from high to low similarity. "Rank-0" denotes not using the retrieved corpus, while "Random" indicates randomly selecting 5 sentences as assistance during the generation process. It can be observed that caption generation performance significantly improves with the top $1 - 5$ retrieved sentences, indicating high-quality relevant corpus. However, as the

**(a) CLEVR-Change**  **(b) Spot-the-Diff**  **(c) Image-Editing-Request**

**GT:** the big green metal object became yellow

**Ours:** the big green shiny ball that is on the left side of the cyan matte thing turned yellow

**GT:** there are more people now

**Ours:** there are three people in the parking lot instead of two

**GT:** take the people out of the back in the photo

**Ours:** remove the people in the background

**GT:** there is no difference

**Ours:** the green cylinder changed its location

**GT:** there are people visible in different sections of the frame

**Ours:** there are two people in the parking lot

**GT:** remove the man from the image

**Ours:** remove the man from the picture

**Figure 6: Case study on (a) CLEVR-Change, (b) Spot-the-Diff, and (c) Image-Editing-Request.**

quality of the retrieved relevant corpus decreases, the performance degrades. Notably, random selection ("Random") even yields worse performance than not using the retrieved corpus ("Rank-0"). These findings underscore the effectiveness of the retrieval-augmented approach in enhancing caption generation, particularly with high-quality retrieved relevant corpus.

## 4.4 Case Study

Figure 6 illustrates six examples by FINER-MLLM across three datasets. Ground-truth change captions and our FINER-MLLM generated change captions are provided for comparison, with changed regions denoted by red boxes. Additionally, the cross-attention distribution between the query features and the image patch features is illustrated, where multiple cross-attention distributions of query features are averaged and presented as a heatmap. Regarding the cases in Figure 6(a), the top case demonstrates FINER-MLLM successfully recognizing the color change of the green ball and providing more detailed location information than the ground-truth caption. While in the bottom case, our method fails to generate the correct change caption, as denoted in red fonts. Further analysis reveals that in this case, the two images are slightly rotated, causing the ground-truth caption to label it as "no difference." However, our FINER-MLLM struggles to recognize these viewpoint changes due to the feature alignment design, instead describing the location change of the green cylinder, which is more concise when ignoring the viewpoint changes. In Figure 6(b), all images are from video-surveillance footage with a fixed viewpoint, and our method successfully captures subtle person changes in both cases. This

can also be observed in Figure 6(c). Moreover, we can intuitively observe from all six heatmaps that attention covers almost every detail within intra-images, with well-aligned attention distributions across paired images. Overall, these observations demonstrate the effectiveness of our designed dual constraints in improving differential-perceptive capabilities and retrieval-augmented caption generation, enhancing generation effectiveness. Moreover, they highlight the real-world application potential of our FINER-MLLM.

## 5 CONCLUSION

In this work, we introduce a differential-perceptive and retrieval-augmented MLLM, enhancing multimodal large language models' ability to discern visual differences across multiple images for change captioning. In particular, we first extracted the image patch features of paired images to facilitate the following differential-perceptive feature extraction. This process incorporates intra-image feature independence and inter-image feature alignment constraints, ensuring that query features capture detailed information and remain aligned with each other, which is crucial for accurately perceiving subtle changes. Additionally, we employed retrieval augmentation to enhance the change caption generation capability of LLM, prioritizing the prediction of relevant words from the retrieval corpus. Extensive experiments have been conducted on three public datasets, and the results demonstrate the effectiveness and real-world application potential of FINER-MLLM. In the future, we plan to extend our method to address tasks involving multiple images, such as video caption, which is an essential problem in multimodal applications.

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
