# OpenReview forum: "Differential-Perceptive and Retrieval-Augmented MLLM for Change Captioning"
_acmmm.org/ACMMM/2024/Conference — MM2024 Poster_

### Official Review · Reviewer_hbnh · 2024-05-23

**Rating:** 5
**Confidence:** 3

**Summary:**

The paper introduces a Multimodal Language Model (MLLM) designed for the change captioning task, which involves describing subtle differences between pairs of similar images.  The key innovation in the proposed model, named FINER-MLLM (diFferentIal-perceptive aNd rEtRieval-augmented MLLM), is the inclusion of a retrieval-augmentation step. This step enhances the model's ability to detect and describe changes by retrieving relevant information from a corpus, which is then used to guide the language model in generating accurate change captions. The effectiveness of FINER-MLLM is demonstrated through extensive experiments on three benchmark datasets: CLEVR-Change, Spot-the-Diff, and Image-Editing-Request. The results show that the proposed model outperforms existing methods, generating more accurate and detailed change captions.

**Strengths:**

* The claim is clear and also well supported through a range of experiments.
* The comprehensive ablation studies effectively illustrate the importance of each architecture component and support the paper.
* The introduction of retrieval in the pipeline is quite novel.

**Limitations:**

* It is unclear whether the retrieval of the top-k words uses an exact index or libraries such as Faiss.
* It would be interesting to see the impact of when varying the number of k retrieved elements
* It would be interesting to see the impact of passing the retrieved tokens concatenated to the input of the LLM instead of using them in the cross attention
* Given that the CIDEr metric has known issues when evaluating captions, particularly with longer and more detailed ones, it would be beneficial to evaluate the proposed approach using more recent metrics like CLIP-S [1] and PAC-S [2].

Minor:
* The authors did not adhere to the review template, and the margins lack line numbers.

Typos:
* The title of Section 3 “Methdology” → “Methodology”


[1] Hessel, Jack et al. Clipscore: A reference-free evaluation metric for image captioning, 2021

[2] Sarto, Sara et al. Positive-augmented contrastive learning for image and video captioning evaluation, 2023

**Suitability:**

3

---

### Official Review · Reviewer_UG2c · 2024-05-24

**Rating:** 4
**Confidence:** 3

**Summary:**

This paper introduces an approach to the task of change captioning, which involves describing subtle changes between pairs of similar images. The authors propose FINER-MLLM, a model that enhances multimodal large language models (MLLMs) with the capability to perceive and describe changes in images. The key innovations include the use of LoRA fine-tuned image encoders for detailed image feature extraction, dual constraints for intra-image feature independence and inter-image feature alignment, and a retrieval-augmented approach to facilitate the generation of accurate change captions. The model was evaluated on three benchmark datasets and demonstrated superior performance compared to existing methods.

**Strengths:**

1. The paper presents a unique integration of differential perception and retrieval augmentation in the context of change captioning, which is a novel contribution to the field.

2. The introduction of dual constraints for feature extraction is technically sound and enhances the model's ability to capture subtle changes.

3. Utilizing an MLLM to accomplish this task represents a promising trend.

**Limitations:**

1. The authors should discuss the performance in general tasks of the fine-tuned MLLM. Does fine-tuning significantly degrade the performance of an MLLM on basic dialogue tasks? The authors can use cases or add experiments to discuss.

2. The improvement in performance appears to be limited, which diminishes the novelty of the paper.

**Suitability:**

3

---

### Official Review · Reviewer_ZvE9 · 2024-05-25

**Rating:** 4
**Confidence:** 2

**Summary:**

This paper introduces FINER-MLLM, a model designed for Change Captioning tasks. It enhances multimodal large language models through refined feature extraction and the integration of retrieval-augmented techniques. By incorporating LoRA fine-tuning and dual constraints, the model effectively captures subtle visual details and ensures alignment across image regions. The adoption of a retrieval-augmented approach further enhances caption accuracy, demonstrating the model's effectiveness in addressing change captioning tasks on benchmark datasets.

Overall, FINER-MLLM presents a comprehensive solution that leverages advanced techniques to improve the generation of change descriptions in image pairs, showcasing promising results in the field of change captioning

**Strengths:**

1, The introduction of FINER-MLLM represents a novel approach in the domain of change captioning by empowering multimodal large language models with differential-perceptive features and retrieval-augmented techniques.

2, The paper provides a comprehensive evaluation of the proposed model on three public datasets, including a synthetic dataset and two real-world domain datasets . By reporting standard metrics such as BLEU-4, METEOR, ROUGE-L, CIDEr-D, and SPICE, the evaluation ensures a fair comparison with existing methods and demonstrates the effectiveness of the proposed approach in generating change captions.

3, The paper is well-structured and clearly articulates the motivation, methodology, experimental settings, and results. The use of figures and examples further enhances the clarity of the presentation, making it easier for readers to understand the concepts and contributions of the research.

**Limitations:**

1, While FINER-MLLM introduces novel aspects such as dual constraints and retrieval-augmented methodology, some individual components like have been utilized in prior works . This may limit the overall novelty of the model in terms of individual components.

2, The effectiveness of the retrieval-augmented approach in generating differential descriptions is contingent on the quality and relevance of the retrieved corpus. In scenarios where the quality of the retrieved corpus is suboptimal, the model's performance may be negatively impacted.

Overall, the paper demonstrates a good work in the field of vision-language understanding and generation. However, concerns arise regarding the novelty and performance parity, particularly in the utilization of components like LoRA fine-tuning and Q-Former, as mentioned earlier.  To enhance understanding of the contribution, it is advisable to provide a clearer explanation of the key insights and technical novelty introduced by the model. This will help highlight the unique approach and innovative aspects of the research. Additionally, for a fair comparison with existing methods, it is recommended to include detailed information on the model settings, backbones, and image resolutions used for performance measurement.

**Suitability:**

3

---

### Official Review · Reviewer_ETa9 · 2024-05-26

**Rating:** 4
**Confidence:** 3

**Summary:**

The paper presents Differential-Perceptive and Retrieval-Augmented MLLM (FINER-MLLM) to enhance the capabilities of multimodal large language models (MLLMs) in perceiving subtle differences and generating accurate change captions. The FINER-MLLM incorporates a fine-tuned image encoder, dual constraints (intra-image feature independence and inter-image feature alignment) for detailed feature extraction, and a retrieval-augmented strategy to aid the generation of differential descriptions. The method is validated through extensive experiments on three benchmark datasets: CLEVR-Change, Spot-the-Diff, and Image-Editing-Request, showing significant improvements over existing methods.

**Strengths:**

1. The integration between differential-perceptive capabilities and retrieval-augmented strategies within MLLMs is a novel approach and has not been explored previously.
2. The dual constraints applied to the feature extraction process (intra-image feature independence and inter-image feature alignment) are theoretically sound and ensure comprehensive extraction of visual details and proper alignment of corresponding features across image pairs.
3. The paper provides extensive experimental validation on three benchmark datasets.

**Limitations:**

1. The method shows inferior performance on the synthetic CLEVR-Change dataset compared to real-world datasets. This limitation is acknowledged in the paper, but a deeper analysis or proposed solution for this discrepancy would be beneficial.
2. The retrieval-augmented approach's performance heavily relies on the quality of the retrieved relevant corpus. This dependency might limit the method's applicability in scenarios where high-quality retrieval data is not available or difficult to obtain.

**Suitability:**

3

---

### Meta-Review · Area_Chair_YRm1 · 2024-06-30

**Recommendation:** Accept (Poster)
**Confidence:** 5

**Metareview:**

The paper initially received for positive indications (1WA, 3BA), which have been confirmed after the rebuttal period, also with a slight increase of the ratings (1BA moved to a 1WA). The paper can be accepted as a poster.